

# Evaluation of Low Impact Development and Nature-Based Solutions for stormwater management: a fully distributed modelling approach

Yangzi Qiu[1], Abdellah Ichiba[2], Igor Da Silva Rocha Paz[2,3], Feihu Chen[4], Pierre-Antoine Versini[1], Daniel Schertzer[1], Ioulia Tchiguirinskaia[1]

[1]HM&Co, École des ponts ParisTech,  Universit éParis-Est, Champs-sur-Marne,77455, France
[2]On leave from École des ponts ParisTech
[3]Instituto Militar de Engenharia, Rio de Janeiro, 22290-270, Brazil
[4]School of Architecture, Hunan University, Changsha, 410082, China

*Correspondence to:* Yangzi Qiu (Yangzi.qiu@enpc.fr)

**Abstract.** Currently, Low Impact Development (LID) and Nature-Based Solutions (NBS) are widely accepted as sustainable approaches for urban stormwater management. However, their complex impacts depend on the urban environmental context as well as the small-scale heterogeneity, which need to be assessed by using the fully distributed hydrological model and high resolution data at small scale. In this paper, a case study (Guyancourt), located in the South-West of Paris, was explored. Three sets of  high resolution X-band radar data were applied to investigate the impact of variability of spatial distribution of rainfall.

High resolution geographic information has been processed to identify the suitable areas that can be covered by the LID/NBS practices, porous pavement, green roof, and rain garden. These individual practices, as well as the combination of the three, were implemented as scenarios in a fully distributed and physically-based Multi-Hydro model, which takes into consideration the variability of the whole catchment at 10 m scale. The performance of LID/NBS scenarios are analysed with two indicators (total runoff volume and peak discharge reduction), with regards to the hydrological response of the original catchment

(baseline scenario). Results are analysed with considering the coupling effect of the variability of spatial distributions of rainfall and land uses. The performance of rain garden scenario is better than scenario of green roof and porous pavement. The most efficient scenario is the combination of the three practices that can reduce total runoff volume up to 51 % and peak discharge up to 53 % in the whole catchment, and the maximum values of the two indictors in three sub-catchments reach to 60 % and 61 % respectively. The results give credence that Multi-Hydro is a promising model for evaluating and quantifying the spatial

variability of hydrological responses of LID/NBS practices, because of considering the heterogeneity of spatial distributions of precipitation and land uses. Potentially, it can guide the decision-making process of the design of LID/NBS practices in urban planning.

# 1 Introduction

Rapid urbanization and climate change are two main reasons that increased the extreme flooding risks in urban area all over

the world (Lovejoy and Schertzer, 2013; Miller and Hutchins, 2017). The adaptation to climate change and prevention of urban





flooding are becoming big challenges in the future (Loukas et al., 2010). Impervious surfaces directly connected with grey infrastructures lead to rainfall transferring into runoff rapidly, which increases the risks of flood events, especially in urban areas (Hollis, 1977). Traditionally, the existing drainage system capacity will be expanded and upgraded to reduce flooding; however, this approach has gradually manifested to be unsustainable, costly, and even unreasonable.

Increasing urban resilience and minimizing the negative impact of urbanization have been emphasized to reduce the risk of urban flood in many countries (Kelman, 2015). Low Impact Development (LID, concept most often used in the U.S.) and Nature-Based Solutions (NBS, concept most often used in the E.U), are two sustainable strategies that are usually used to reduce the influences of human activities on natural environment, especially for stormwater management. LID emphasizes using a suite of small-scale controlled practices, including rain garden, green roof, bioretention swale, porous pavement, and

rainwater tank, to conserve or recover the natural environment of a region (Newcomer et al., 2014). These have been implemented in the last several years to decrease the risk of urban flood and increase the resilience of city (Dietz, 2007; van den Bosch and Ode Sang, 2017). NBS is defined by the European Commission, which are actions inspired by, supported by or copied from nature (European Commission, 2015). To some extent, NBS concept builds on, and supports, LID, blue and green infrastructure (B&GI), and other closely related concepts (Berry et al., 2015). The Climate-KIC Innovation project Blue

Green Dream focuses on the use of Blue-Green Solutions to achieve urban sustainability and climate change resilience, which has demonstrated NBS can provide cost-effective, sustainable enhancements to quality of life, and resilience to extreme weather events (Bozovic et al., 2017). NBS provides multiple benefits to an urban area, such as combatting the effects of heat island, conservation of biodiversity, and bringing varied natural features into cities, by means of adaptation to local conditions, resource-efficient and systemic interventions (Cohen-Shacham et al., 2016).

The performance of LID/NBS practices have been analyzed in terms of the reduction of runoff volume and peak flow in different urban catchments (Ahiablame and Shakya, 2016; Hu et al., 2017; Bloorchian et al., 2016; Zahmatkesh et al., 2015), and some researchers found that the properties of LID practices and rainfall characters will have an influence on their performance (Martin-Mikle et al., 2015; Seo et al., 2017; Holman-Dodds et al., 2003). In addition, with the purpose of simulating the hydrological response of LID/NBS practices, several hydrological models were developed, such as Storm Water

Management Model (SWMM), Long-Term Hydrologic Impact Assessment-Low Impact Development (L-THIA-LID), Soil and Water Assessment Tool (SWAT) etc. SWMM is one of the frequently used semi-distributed hydrological models to investigate the impact of LID/NBS practices on urban runoff and water quality (Palla and Gnecco, 2015; Sun et al., 2014; Cipolla et al., 2016; Jia et al., 2015; Kong et al., 2017; Kwak et al., 2016). Liu et al., (2015) used L-THIA-LID model to analyze the differences of watershed responses with the change of land use. With SWAT, the influencing factors related to the

performance of LID were summarized into three aspects, including the configurations, application areas, and characteristics of rainfall event (Her et al., 2017). Brunetti et al., (2017) developed surrogate modeling, which was used for numerical analysis of the hydraulic behavior of LID techniques.

As mentioned, a given number of practical studies investigated the performance of LID/NBS practices on the reduction of urban flood risks (Massoudieh et al., 2017; Seo et al., 2017). However, most studies on modeling the hydrological impacts of



LID/NBS practices are based on semi-distributed hydrological model and design storm in comparatively small watersheds (Qin et al., 2013; Peng et al., 2019; Zhu et al., 2019). Burszta-Adamiak and Mrowiec, (2013) demonstrated that the hydrological performance of SWMM is not very explicit for the study area with only pervious and impervious land covers. Rossman et al. (2010) also indicated SWMM has some limitations in reflecting complicated urban catchment, which is difficult to show different hydrological responses to a variety of urban land uses. Versini et al. (2018) and Gilroy and McCuen, (2009) adopted

either fully distributed or spatio-temporal hydrological model to analyse the impact of different NBS practices, but in any case using only homogeneous design storms. Therefore, the influence of the heterogeneity of precipitation distribution were not considered in these studies. In general, rare research used the high resolution rainfall data (i.e., X-band radar data with 250 m spatial resolution) and considered the coupling effect of the spatial distributions of precipitation and land uses. Therefore, the performance of LID/NBS practices in stormwater management still need to be further analyzed with a fully distributed model

and high resolution rainfall data at an urban catchment because the model is able to consider heterogeneous surfaces at very small scales.

In this study, a fully distributed and physically-based hydrological model (Multi-Hydro) is initially applied to investigate the hydrological impacts of different LID/NBS practices on a peri-urban catchment at 10 m scale by using high resolution rainfall data from X-band radar. The performance of four LID/NBS scenarios is evaluated and compared with baseline scenario in

terms of discharge in the drainage system under three distributed rainfall events, considering the coupling effects of variability of spatial distributions of precipitation and land uses. Section 2 presents the data preparation process. It is worthy to mention that rainfall events are sufficiently long, hence potentially could be subdivided into shorter ones with quite different behaviors, which can enrich the rainfall sampling. The results are analysed with two indictors (peak discharge reduction and total runoff volume reduction) in the whole catchment and in the three sub-catchments. Through the comparison and evaluation, the

complexity of hydrological behaviors regarding to coupling multi-factors is highlighted and discussed in section 3.

## 2 Materials and methods

### 2.1 Study area

This case study is conducted on the city of Guyancourt in France, a peri-urban catchment located in the southwest suburbs of Paris at the Saclay Plateau, as shown in Fig. 1, is expected to become "French Silicon Valley". The total area of the catchment

is around 5.2 km$^2$, with a population of about 30,000. Based on the land use map provided by the French National Institute of Forest and Geographic Information (IGN), the urbanized degree of the catchment area is 37.6 %. This area has an oceanic climate, and the average annual temperature and precipitation is 10.7 °C and 695 millimeters (Météo-France). Generally, it has a steady reliable rainfall throughout the whole year. However, due to the climate change, a growing number of extreme flood events have happened in this region. Especially in recent years, a large amount of surface runoff and a higher peak flow

rate were caused due to the increase of impervious land cover. Thus, a substantial number of surface runoff may go beyond the capability of drainage system and some low-lying areas in the catchment will suffer more easily from waterlogging. Here,



a map with a 25 m resolution Digital Elevation Model (DEM) was created by a Geographic Information System (GIS) tool to find low-lying areas at risk of waterlogging in the catchment, shown in Fig. 1. The blue spots with a total area of 0.6 km$^2$ in this map represent the low-lying areas that are easily flooded by stormwater. Similarly, the yellow spots indicate the vulnerable buildings that lie within and adjacent to them.

## 2.2 Data preparation

The rainfall data was derived from the polarimetric X-band radar, located in École des ponts ParisTech (ENPC), Champs-sur-Marne. The distance between the X-band radar and the catchment is around 50 km. The spatial resolution and temporal resolution of the X-band radar are 250 m and 3.4 min, respectively. Three relatively long rainfall events with different characteristics that happened in the year of 2015 were chosen for the study. The first event occurred on 12th and 13th September 2015 with duration of 44 hours, starting from 04:05 and ending at 00:00. This event was caused by two consecutive depressions coming from North-West of British, being affected by the Cévenol events simultaneously. Storm Henry has an influence on the second event that occurred on 16th September 2015, lasting 16.8 hours, starting from 00:05 and ending at 16:50, and was ascribed to a tropical depression from the South, resulting in strong winds. Considering the first 8.4 hours of the second event has little rainfall, only the 8.4 remaining hours were taken into consideration in this study. The third event occurred on 5th and 6th October 2015, lasting 31 hours, starting from 09:10 and ending at 16:05, and with the influence of a depression from the West (Paz et al., 2018) (see Table 1 for more details). Figure 2a shows the maps of cumulative rainfall by radar pixels, which indicates that the rainfall is very variable in the whole catchment. The maximum cumulative rainfall by each radar pixel in the three events are 36.9, 14.1, and 25.4 mm, respectively. Figure 2b presents the time evolution of rainfall rate and cumulative rainfall of three events over the whole catchment. As mentioned, the three events are sufficiently long which can be subdivided into several portions, for the event of 12/09/2015, the highest rainfall peak reached 20 mm/h, and the rainfall accumulate very fast around the first and third rainfall peak. The maximum rainfall rates of the second and third rainfall event are 9 mm/h and 36.4 mm/h respectively. Although the highest rainfall peak of the third event is 36.4 mm/h, it only lasts for 3 min, which has a small influence on the cumulative rainfall.

The land use map and topography of the catchment are distributed data (usually with a GIS format) that must be represented with a high resolution because the data quality and resolution is significant to the sensitivity of Multi-Hydro model. The initial GIS data preparation process was done by Neto et al., (2018). The original land use was consists of six types, including forest, grass, road, building, gully, and water. Here, the original land use data were used to compare with satellite images from Google Maps. Then, a new land use type, parking, was added in the catchment. In total, seven land use types occupied 28.8 %, 32.7 %, 9.6 %, 15.5 %, 1.9 %, 0.9 %, and 10.6 % of the total area respectively, as shown in Fig. 3. The pervious surface accounts for 62.4 % of total area, and the corresponding impervious surface is 37.6 %.

The topography is presented as a 25 m resolution DEM, which was also obtained from the IGN. The whole catchment is relatively flat. As shown in Fig. 4, the altitude in the North is slightly higher than that of the South. The highest altitude in the whole catchment is 175.1 m and the lowest altitude is 143.39 m in the Etang des roussière pond.



The data of drainage system was obtained from La préfecture de Saint-Quentin-en-Yveline. In the catchment, the drainage was designed with a capacity of 10-year return period, and it consists of 4,474 nodes and 4,534 conduits, with a total length of 76 km.

Soil data for the catchment was obtained from the InfroTerre Database (http://infoterre.brgm.fr). As shown in Fig. 3, the soil data of three points are selected, which indicates that sand clay is located at the first layer of the soil profile. For point 1, there

exists a layer of limestone soil, which is less permeable. Silver sand which has the best infiltration ability, was in the next layer. The last layer is silver sand and clay, which has better infiltration ability than limestone. From a hydrological point of view, the soil data shows the complexity of the subsurface of the catchment. Therefore, the soil profile is reasonably simplified into three layers: sand clay layer (0-10.5 m), silver sand layer (10.5-25.8 m) and silver sand and clay layer (25.8-40 m).

### 2.3 Multi-Hydro model

The Multi-Hydro model was developed by ENPC, which is a fully distributed and physically based hydrological model used for assessing hydrological response at the urban scale (Ichiba et al., 2018; Giangola-Murzyn, 2013; El Tabach et al. 2009). It is an interaction core that utilizes four open source modules (surface module, rainfall module, drainage module, and infiltration module). Each of them can represent a main part of the hydrological cycle, as shown in Fig. 5.

The Two-Dimensional Runoff Erosion and Export (TREX) model consists of the surface module (MHSC), which aims to

model surface runoff and infiltration at each pixel, depending on the land use categorization (roads, forests, houses, and water)(Velleux et al., 2008). The diffusive wave approximation of Saint-Venant equations are used for calculating the overland flow, following the conservation of mass and momentum equations. The rainfall module can deal with different kinds of rainfall data (from radar or rain gauge). The Multifractal framework is used for downscaling the rainfall input needed for the Multi-Hydro model (Schertzer and Lovejoy, 1987; Lovejoy and Schertzer, 1990), and has been widely used in geophysical

fields with an extensive ranges of scales (Lovejoy and Schertzer, 2011). More details about downscaling processes can be found in Gires et al., (2013). To simulate the sewer network, 1D SWMM model was used as the drainage module (MHDC) in Multi-hydro (James et al., 2010), which represents the flow computed by 1D Saint-Venant equations in conduits and nodes. The infiltration module (MHGC) is based on the Variably Saturated and 2-Dimensional Transport (VS2DT) model, which is capable of simulating the process of infiltration in the unsaturated subsurface zone. Besides the main modules, several

'LID/NBS infrastructures', including green roof, basins and barriers, are comprised in the latest version of Multi-Hydro (more details in Versini et al., 2016).

### 2.4 Simulation scenarios

According to the catchment conditions and the land development requirements, five simulation scenarios were proposed in this study, covering baseline, porous pavement, rain garden, green roof, and the combined scenario. They are explored to

access the performance of the LID/NBS implementation under the three rainfall events. These scenarios are described below.





The baseline scenario is considered as a base case in the catchment without any LID/NBS practices. The capability of drainage system was designed to resist a 10-year return period storm event.

For the porous pavement scenario (as shown in Fig. 6a), one of the land use types, road, is divided into two categories, roadway and walkway, with a width between 1m to 7.5 m. Considering the vehicle load, busy roads were supposed not to be reasonably

replaced by porous pavement. Therefore, porous pavement was hypothetically implemented on the non-driveways (width equal and less than 2.5 m) and all parking lots. In the catchment, only 46 roads satisfy the condition. Finally, 14.5 % of the whole area is appropriate to be replaced by porous pavement.

For rain garden scenario (Fig. 6b), the low elevation greenbelts around houses were implemented by rain gardens, which can collect and store up the surface runoff from surrounding impermeable areas before infiltration on site. When rain garden

saturated, the redundant surface runoff will drain into the drainage system. On the basis of application condition of rain gardens and the urban planning of the city of Guyancourt, 11.5 % of the whole area is set as rain gardens in the catchment.

In the catchment, most of the buildings are houses with sloped roofs. Other types of buildings with flat roofs, only constitute one third of the total building area. According to the properties of green roof, small and light green roofs consisting of a soil layer and a storage layer are implemented on all flat ones, which can be simulated bythe green roof module. All slope roofs

remain unchanged. Finally, green roofs were applied to 11.5 % of the whole area (Fig. 6c).

The combined scenario (see Fig. 6d), combined the three aforementioned LID/NBS practices. Those practices occupy 37.5 % of the whole catchment. In this case, the area of pervious surface reached 4.6 km$^2$, which is about 88.4 % of the whole catchment.

**2.5 Methodological framework**

The general hydrological response of five scenarios under three rainfall events has been assessed by Multi-Hydro model and its green roof module ($3\times5$ simulations). Two index ($\Delta$V, total runoff volume reduction; $\Delta$Qp, peak discharge reduction) are calculated:

$$\Delta Qp(\%) = \frac{Qp_0 - Qp_i}{Qp_0} \times 100 \tag{1}$$

$$\Delta V(\%) = \frac{V_0 - V_i}{V_0} \times 100 \tag{2}$$

where $Qp_0$ refers to peak discharge and $V_0$ represents total runoff volume in baseline scenario, and the $Qp_i$ and $V_i$ are peak discharge and total runoff volume of other scenarios with different LID/NBS.

**2.6 Modelling setup**

For the implementation of Multi-Hydro, the first step is to change land uses from vector to raster. During this process, the input files of the DEM and land use map were put into the input data generator - Multi-Hydro Assimtool (MH-A). Then, a

unique land use class will specify the hydrological and physical properties, which is assigned to each pixel. Here, in order to attribute land use to a unique class, two methods were used (See Fig. 7). The first one is the priority rule defined by users. For



the purpose of keeping the continuity of the roads, this rule assumes that the impervious land uses have a higher priority than the pervious, which will overestimate the proportion of impervious land use. The other one is majority rule proposed by Ichiba et al., (2016). The main idea of this rule is to use very small pixels (low to 20 cm), and to attribute the land use on the basis of the main land use type. However, it is necessary to follow the priority rule for gully so as to connect surface module and drainage module. Finally, the impervious land uses with priority rule occupied 54 % of the whole catchment, which is 14 % higher than that of majority rule. The second step is to define the soil parameters. In Multi-Hydro, there are four soil layers, at most, that can be defined. In this study, three layers were adopted as the aforementioned. The last step is to input the data of the drainage system into MH-A, which contains the details of all the system components (inlet and outlet nodes, geometry, length, as well as diameter). Based on the fully distributed characters of Multi-Hydro model, users can choose a specific spatial resolution. In this study, after all data were pre-treated by MH-A, Multi-Hydro was implemented to simulate the hydrological response of each scenario with a 10 m spatial resolution (the grid system will create square grids with a cell size of 10 m), and 3 min temporal resolution.

All the model parameters related to the land use type and soil type were selected from the Multi-Hydro model manual (Giangola-Murzyn et al., 2014), as shown in Table 2 and Table 3.Green roof is a special LID/NBS practice which needs to be simulated with the Multi-Hydro green roof module. The properties of the green roof are illustrated in Table 4.

Before LID/NBS scenario simulation, the Multi-Hydro model was validated in terms of the water level of Etang des roussière in three rainfall events (12/09/2015, 16/09/2015, and 05/10/2015). The model performance is evaluated through two indicators, Nash-Sutcliffe Efficiency and relative error (RE). The Nash-Sutcliffe Efficiency (NSE $\in$(-∞, 1]) is an indicator generally used to verify the quality of the hydrological model simulation results, the relevant equation is described as follows:

$$\text{NSE}(A_i, B_i) = 1 - \frac{\sum_{i=1}^{n}(B_i - A_i)^2}{\sum_{i=1}^{n}(B_i - \overline{B_i})^2} \tag{3}$$

where $A_i$ refers to simulated values, $B_i$ refers to observed values, and $\overline{B_i}$ represents the total of average observed value. The Nash is closer to 1, indicating that the model is more reliable. Nash is closer to 0, indicating that the simulation result is closer to the average observed value, which means the result is credible, but the simulation error is larger. If NSE is far less than 0, it means the result is unauthentic.

The relative error (RE) represents the difference between observed values and simulation values, which reflects the reliability of the simulation values.

$$\text{RE}(A_i, B_i) = \frac{\sum_{i=1}^{n}|B_i - A_i|}{\sum_{i=1}^{n}B_i} \times 100 \tag{4}$$





# 3 Results and discussion

## 3.1 Model validation

With respect to the observed and simulated water levels in the baseline scenario, the NSE coefficients and RE values are presented in Table 5. The NSE coefficients of priority rule and majority rule of three events are higher than 0.9. Figure 8 revels the time evolution of the observed and simulated water levels in three rainfall events. On the whole, the simulation values are matched with the observed values, which indicates the Multi-Hydro model is reliable to reflect the variation of water levels, but with a certain deviation. The relative error values of event 12/09/2015 and event 16/09/2015 with priority rule are lower

than 10 %, but that of the event 05/10/2015 is 15 %, because the model underestimated the water level at the third rainfall peak. The relative error of majority rule is higher than that of priority rule due to the majority rule results in a larger drop of water level. Finally, the LID/NBS scenarios are simulated with the priority rule.

## 3.2 General effect of LID/NBS scenarios on the whole catchment

For evaluating the hydrological response of each scenario in the whole catchment, the sum of the discharge in four conduits (highlighted in Fig. 9) that eventually merge into the storage unit of the drainage system were considered. Because the three distributed rainfall events have different durations, intensities, and amounts, the discharge of the sum of the four conduits in each rainfall event has a significant difference. Considering the rainfall amount of each event, the event of 12/09/2015 is the strongest one, followed by 05/10/2015 (moderate) and 16/09/2015 (weakest). Therefore, the discharge of baseline scenario

reaches 8.47 $m^3$/s in strongest event, which is around two times as higher as the moderate or weakest event. In addition, three hydrographs that show very similar trends are illustrated in Fig. 10. Compared with the baseline scenario, the LID/NBS scenarios can decrease the discharge in varying degrees, especially efficiently in the first rainfall peak. As shown in Fig. 10c, the moderate event was composed by several periods, for the first period, the discharge of porous pavement, rain garden, green roof, and the combined scenario in the first rainfall peak, is less than the baseline scenario, 0.5, 0.6, 1.0 and 1.2 $m^3$/s respectively.

Conversely, for the second rainfall peak, the difference in discharge between LID/NBS scenarios and baseline scenario is gradually diminished. The others two rainfall events show similar results. Here, it is needs to be mentioned that all of LID/NBS practices were considered as unsaturated at the beginning of the rainfall events in the simulation. In this case, during the first portion of the rainfall event, the rainfall can quickly infiltrate through LID/NBS practices or be stored in the substrate of LID/NBS practices, which will reduce the first corresponded peak discharge efficiently. However, the substrate of LID/NBS

practices will be saturated after a long period with strong rainfall. If LID/NBS practices have no enough condition to recover its retention capacity, at the beginning of next portion of the rainfall event (e.g., the third portion of 12/Sep/2019 event), the LID/NBS scenarios produce higher discharges than that of baseline scenario.





### 3.3 General effect of LID/NBS scenarios on sub-catchments

According to the distribution of the drainage system, the catchment can be divided into three sub-catchments: SUB1, SUB2
and SUB3 (Fig. 9). It is important to note that the discharge of the sub-catchment SUB1 will flow into the conduits 4541 and
4542, while the discharge of SUB2 will flow into conduit 4543 and the one of SUB3 into 4544. Figure 11 revels the nine
hydrographs in three sub-catchments under three rainfall events. The hydrographs show that the discharge of baseline scenario
is higher than that of four LID/NBS scenarios. And these discrepancies in SUB1 and SUB2 are much more obvious, which are
related to the spatial distributions of the drainage system, land uses, and precipitation (i.e., with different intensities in each
sub-catchment under the same rainfall event, especially the peak of rainfall). More discussions are shown in section 3.5.

For three rainfall events, the maximum discharge of baseline scenario in SUB2 is only 0.25 m$^3$/s (SUB1: 7 m$^3$/s, SUB3: 1.2
m$^3$/s), while that of the maximum in four LID/NBS scenarios decreased to 0.02 m$^3$/s (SUB1: 1 m$^3$/s, SUB3: 0.4 m$^3$/s). The
results demonstrate that only a small part of surface runoff entered into the drainage system of SUB2, while most runoff
infiltrated into the subsurface or flowed into another sub-catchment. On one hand, the topography of SUB2 is higher in the
North and lower in the South, and a large forest accounts for one third of total area of SUB2, located in the South. Therefore,
the surface runoff can move into the forest and infiltrate more efficiently. On the other hand, the distribution of the gullies is
relatively sparse in SUB2, which limited the runoff flow into the drainage system. On the whole, the hydrographs of sub-
catchments present the fully distributed hydrological model Multi-Hydro can sensitively reflect the heterogeneity of the urban
patterns and the spatial variability of the precipitation.

### 265    3.4 Evaluation of LID/NBS scenarios in the whole catchment

In order to calculate ΔQp and ΔV of the whole catchment under four scenarios, Eq. (1) and Eq. (2) were used. As shown in
Fig. 12, for porous pavement scenario and rain garden scenario, peak discharge reduction is more evident than total runoff
volume reduction in three rainfall events, where the average ΔQp is around 6 % higher than average ΔV. The reason is that the
three rainfall events have relatively long and continues rainfall, the retention capacity of porous pavement and rain garden will
be reduced with the continuing rain. Furthermore, the ΔQp of the two scenarios in the weakest event are lower than that of two
stronger events. In this case, porous pavement and rain garden are efficient to attenuate the peak discharge, especially for the
high intensity peak of rainfall. Similarly, ΔV of the two scenarios in moderate event is the highest. For the porous pavement
scenario, ΔV of the moderate event is about 5.5 % and 7.9 % higher than ΔV of the weakest event and strongest event,
respectively. For the rain garden scenario, ΔV of moderate event is 7.3 % higher than that of weakest event and 11.8 % higher
than that of strongest event. It is mainly because the three rainfall events have different durations and intensities. The weakest
event has 8.4 hours continues rainfall in total, while the moderate event has some dry periods can let porous pavement and rain
garden recover its retention capacity. Especially, the first portion of moderate event lasts about 7 hours with low rainfall
intensity (less than 5 mm/h). During this period, porous pavement and rain garden did not reach their limitations, and after 10
hours dry period, their retention capacity recovered. However, although the strongest event also has some dry periods, the


average rainfall intensity of strongest event is twice as higher as the two other events. On the whole, it is concluded that the porous pavement and rain garden have better performance with intermittent and sometimes intense rainfalls.

Compared with the first two scenarios, Green roof scenario has better performance than porous pavement scenario in terms of ΔQp and ΔV, but slightly worse than rain garden scenario. The average ΔQp and ΔV of green roof scenario is 23 % and 20 %, respectively. For total runoff volume reduction, green roof scenario is also more effective in moderate event and less effective

in strongest event. ΔV of green roof scenario in the strongest event is less 10.1 % and 11.4 % than that of the weakest event and moderate event. As shown in Fig. 10, at the beginning of the rainfall events, green roof seems to be particularly efficient than the other LID/NBS practice, but after the first rainfall peak, more discharge will be generated in green roof scenario. The reason is related to the fact that the initial unsaturated substrate of green roof with a thickness of 0.03 m can quickly store the precipitation at the beginning of the rainfall event, then with continuous strong rainfall, the substrate of green roof will get

saturated gradually, finally more water will become the runoff and flow into the drainage system. Generally, green roof has better performance in short and weak rainfall event in terms of total runoff volume reduction, but more effective to attenuate the peak discharge in the rainfall event with higher intensity. Some similar results were concluded in various previous studies (Carter and Rasmussen, 2007; Hakimdavar et al., 2014)

In the combined scenario, all kinds of the LID/NBS practices was implemented on the catchment. Evidently, it is the most

effective configuration due to almost one third of the catchment occupied by LID/NBS practices. ΔQp and ΔV are about 10-20 % higher than that of LID/NBS scenarios with single practice. The maximum ΔQp and ΔV reached 53.7 % and 51.9 %, respectively. However, only one particular example is the moderate event, in which ΔQp of rain garden scenario is higher than that of combined scenario. As shown in Fig. 10c, at the second rainfall peak of moderate event, it generated about 0.1 m³/s discharge in combine scenario more than that in rain garden scenario. The reason is that combined scenario was affected by

each LID/NBS practice. If any kind of LID/NBS practice was filled up during a rainfall event, for example, green roofs or porous pavements, it will reduce the ΔQp of combined scenario. Peng et al., (2019) also reported that with higher rainfall intensity, the performance of combined LID scenarios is less effective than single practice scenario, because some practices in the combined scenario have no enough detention storage capacity.

**3.5 Evaluation of LID/NBS scenarios in sub-catchments**

In order to calculate the final discharge of SUB1, the sum of discharge in conduits 4541 and 4542 was taken into consideration. As illustrated in Fig. 13, the peak discharge reductions with different LID/NBS scenarios and rainfall events in three sub-catchments varied between -13 % to 61 %, and total runoff volume reduction varied from 3 % to 60 %. The hydrological impact of LID/NBS scenarios is significantly different in each sub-catchment, this can be explained by each kind of LID/NBS practice has a different proportion in the sub-catchments and by the fact that the rainfall is distributed unevenly in the whole

catchment (see Fig. 11 and Table 6 for more details).

In the three rainfall events, the order of ΔQp of LID/NBS scenarios in SUB1 from high to low is: combined, rain garden, green roof and porous pavement (Fig. 13a, c, e). Compared to the scenarios of green roof and porous pavement, rain garden scenario





has the highest ΔQp, though rain garden practices only occupy 5 % in SUB1. Apparently, the ΔQp of LID/NBS scenarios has a non-linear relationship with the area of LID/NBS practices in the sub-catchment. Because the decrease of peak discharge

due to the application of single kind of LID/NBS practice is not only related to its area, but also its capacity. For example, the rain garden practice has higher storage capacity than that of practice of green roof and porous pavement. When the precipitation exceeds the storage limit of LID/NBS practices, the exceed rainfall will form surface runoff. That can explain why green roof practices account for around 10 % in SUB2 but with the lowest ΔQp. For the ΔV of LID/NBS scenarios in the weakest event, green roof scenario in SUB1 is around 30 %, which is the same as the rain garden scenario and 11 % higher than that of porous

pavement. Because green roof practices account for 10.9 % in SUB1, which are 2 % and 5 % higher than practices of porous pavement and rain garden respectively. In addition, the average cumulative rainfall of SUB1 is around 2 mm lower than that of the other sub-catchments.

For the SUB2, the ΔQp of rain garden scenario in the two stronger events are around 2 % higher than that of combined scenario. As mentioned, the performance of combined scenario was affected by each kind of LID/NBS practice. In the case, when any

individual solution has negative impact, the performance of the combined scenario will weaken. In the SUB3, the ΔQp of scenario of porous pavement and combined in the moderate event are negative. According to the hydrograph (Fig. 11i), the discharge of porous pavement scenario is 0.1 m³/s higher than that of baseline scenario in the strongest rainfall peak (36 mm/h), which demonstrates porous pavement practices reached their infiltration capacity at this rainfall peak. Some similar results are concluded that green roof can generate a peak discharge higher than that of the current situation in Carter and Rasmussen,

(2007). Due to this reason, porous pavement practices generate a negative influence on combined scenario as discussed previously. The ΔQp and ΔV of the SUB3 are obviously lower than that of the other two sub-catchments in the three rainfall events. This is partly because the average cumulative rainfall of SUB3 is stronger than that of the two others sub-catchment. In addition, the proportion of the total NBS area in SUB3 is lower than that of the SUB1 and SUB2, and the drainage system is distributed intensively in SUB3. Therefore, the runoff drained into the drainage system more quickly.

To some extent, the implementation of LID/NBS practices can lead to a significant reduction of the stormwater quantity reaching the drainage system and decrease the risks of flood in the catchment. Firstly, the performance of those scenarios is strongly related to rainfall characteristics and distributions, the substrate of LID/NBS practices are more easily saturated under the rainfall with high intensity and long period. Secondly, their performance also depends on the properties and spatial distribution of the LID/NBS practices. Compared with the other two scenarios with single practice in each rainfall event, rain

garden scenario is the most efficient one to reduce total runoff volume and peak discharge, followed by green roof, and porous pavement. The reason may lie in the fact that rain garden is the low elevation greenbelt and simulated with maximum depth 0.3 m, which reflects its storage capacity is better than that of other LID/NBS practices. Dussaillant et al., (2004) tested the depression depth of rain garden with 10, 15 and 45 cm, which demonstrated that increasing the storage depth will decrease the total saturation time. Qin et al., (2013) also indicated that increasing storage capacity of LIDs can improve the performance of

the system effectively. In this study, the ΔQp of all LID/NBS scenarios are relatively more effective in strongest event. In addition, the ΔV of LID/NBS practices are highest in the moderate event. It is related to the characteristics of the rainfall events.





As mentioned earlier, the moderate event can be divided into several portions, the first portion has low rainfall intensity and there is a relatively long dry period between the first portion and the second portion. In this case, more rainfall runoff can infiltrate into underground through LID/NBS practices or be stored in the substrate of LID/NBS practices. On the contrary,

the weakest event has a relatively longer continuous rainfall, and more water is finally drained into the drainage system. Some similar results were well documented (Pyke et al., 2011; Palla and Gnecco, 2015). Qin et al., (2013) concluded permeable pavement, green roof, and swale can effectively deal with urban runoff during heavier and shorter storm events.

Obviously, any single practice is efficient, particularly, rain garden. Nevertheless, the overall performance will improve when they are combined, as the combined scenario shows. Ahiablame and Shakya, (2016) also found the combination of two or

three practices are more useful to reduce substantial flood risks. Hu et al., (2017) represented the most effective solution is combination of cistern and permeable pavement, which can reduce 80 % of a flood hazard level. The selection of LID/NBS practices in a certain area is strongly dependent on the site conditions, targets and potential funds etc., which should be chosen carefully to archive the goal of stormwater management. In this study area, among three single practices, rain garden shows the best performance with two advantages: (1) a better landscape effect; (2) easily implemented by house owners, which can

be a recommended solution. By comparison, green roof and porous pavement need more conditions to implement. Considering the target of stormwater management, rain garden is the dominant practice as a retrofit technology without considering economic spending.

## 4 Conclusions and perspectives

This paper studies the hydrological impacts of LID/NBS practices on a semi-urban catchment during three distributed rainfall

events with a fully distributed and physically-based hydrological model (Multi-Hydro). The high resolution distributed rainfall data from X-band radar reflect the spatial and temporal variability of precipitation. Multi-Hydro model fully considered the extensive complicacy of an urban environment at small scale, including the location of LID/NBS practices and spatio-temporal distribution of precipitation. The major findings are summarized as follows:

1.    The results illustrate that implement LID/NBS practices can significantly reduce the urban runoff. Although the three

370        rainfall events have different characteristics (amount, duration, and intensity), every LID/NBS scenario is more effective at the beginning of the rainfall event. After a certain period of rainfall, the substrate of LID/NBS practices reaches saturation, the discharge in the conduits of drainage system of LID/NBS scenarios will increase, and then the gap of discharge between LID/NBS scenarios and baseline scenario will narrow, which point out that LID/NBS scenarios are better to reply on short but intense rainfalls. Multi-Hydro give more quantitative insight on this saturation process and

375        demonstrates that a combination of the three LID/NBS practices seems to be the optimal solution to significantly absorb the runoff before flowing into the drainage system during different rainfall events, and to fulfill regulations established by local stormwater managers. The total runoff volume reduction and peak discharge reduction of combined scenario can reach more than 50 % in the whole catchment, and more than 60 % in the sub-catchment with respect to the 6 months or





2-5 years return period rainfall events. NBS practices could be very useful to avoid some waterlogging in Guyancourt

catchment or in Paris region.

2.  In the whole catchment, each LID/NBS scenario is more effective in two stronger but short events. The performance of rain garden scenario is better than the other two scenarios with single practices, and rain garden scenario has a steady performance in various rainfall events. In the three events, rain garden scenario can reduce 26.2 % of total runoff volume and 31.4 % of peak discharge averagely. Green roof scenario shows better performance than porous pavement scenario.

However, implementation of green roof is strongly related to the roof conditions, and it incurs a higher cost. Porous pavement scenario can averagely reduce the total runoff volume and peak discharge around 16 % and 21 %, which can be considered in some parking lots and footpaths. Considering the comprehensive conditions, rain garden is the easiest practice to deploy in this catchment. But in general, it is necessary to combine three kinds of LID/NBS practices to handle the urban waterlogging at a large scale.

3.  In the sub-catchments, the significantly different hydrological responses of LID/NBS scenarios indicate that their performance is influenced by the coupling effect of variability of spatial distributions of precipitation and land uses (e.g., the rainfall amount, rainfall intensity, proportion of LID/NBS practice). The coupling effect induced by these factors are reflected in two aspects: (1) the discrepancy of maximum discharge of three sub-catchments reach 20 times in the same scenario and the same rainfall event; (2) the peak discharge reductions with different LID/NBS scenarios and rainfall

events in three sub-catchments varied between -13 % to 61 %, and total runoff volume reduction varied from 3 % to 60 %. Generally, the performance of rain garden scenario is highlighted by the results for all the sub-catchments, rain garden scenario is always performing better than the other single scenarios. However, the impact of other single scenarios, like porous pavement and green roof, rather depend on sub-catchment conditions and/or rainfall event. From the design point of view, the choice of rain gardens has a further advantage of offering a multi-functional use of the space.

In this study, the presented results appear optimistic because the LID/NBS practices can be a sustainable approach to retrofit the waterlogging and promote the environmental quality of urban area. The analysis highlights the complexity of hydrological response with respect to the urban context and the spatial distribution of the precipitation at small scales. The fully distributed structure of Multi-Hydro model was demonstrated suitable for evaluating LID/NBS performance, which can be as a decision support tool to give certain information to decision-makers for urban planning and choose an optimal plan of LID/NBS

practices. It is a new development in hydrological research approach in LID/NBS techniques. At present, Multi-Hydro model has been developing. In the future, more LID/NBS practice modules will be added, and among these, different combinations and configurations can be investigated, such as green roofs with different substrate porosities or thicknesses, or different ratio of LID/NBS practices.

*Data availability*. The simulation input data and the simulated discharges of the five scenarios investigated in this paper can be accessed through: doi: 10.5281/zenodo.3270816.

.





*Author contribution.* Yangzi Qiu prepared the manuscript with contributions from all co-authors. Ioulia Tchiguirinskaia and
Daniel Schertzer conceived and designed the study. Yangzi Qiu and Igor Paz prepared the input data for the simulations.
Abdellha Ichiba developed the model code. Yangzi Qiu performed the simulations. All co-authors revised the manuscript.

*Competing interests.* The authors declare that they have no conflict of interest.

## 5 Acknowledgement

Yangzi Qiu greatly acknowledges the financial support by the China Scholarship Council as well as enlightening discussions
with Hao Wang and Marty Randall. The authors are thankful to the Chair "Hydrology for Resilient Cities" (endowed by
VEOLIA), for partial financial supports. The authors also thank SIAVB and Veolia, particularly Bernard WILLINGER, for
providing data and for useful discussions.

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

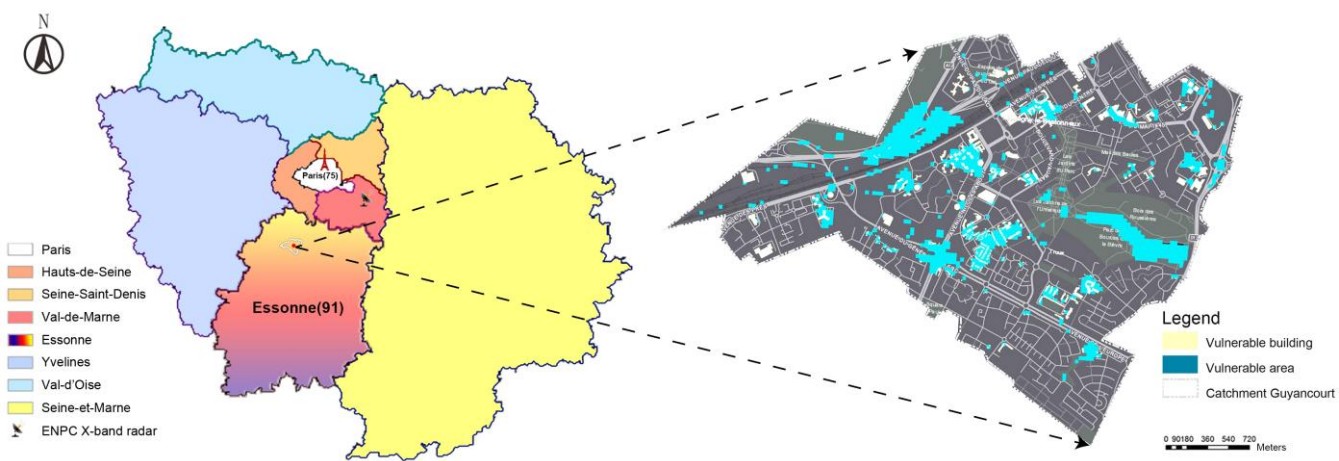


**Figure 1: Study area of Guyancourt in France and the map of vulnerable area to waterlogging.**





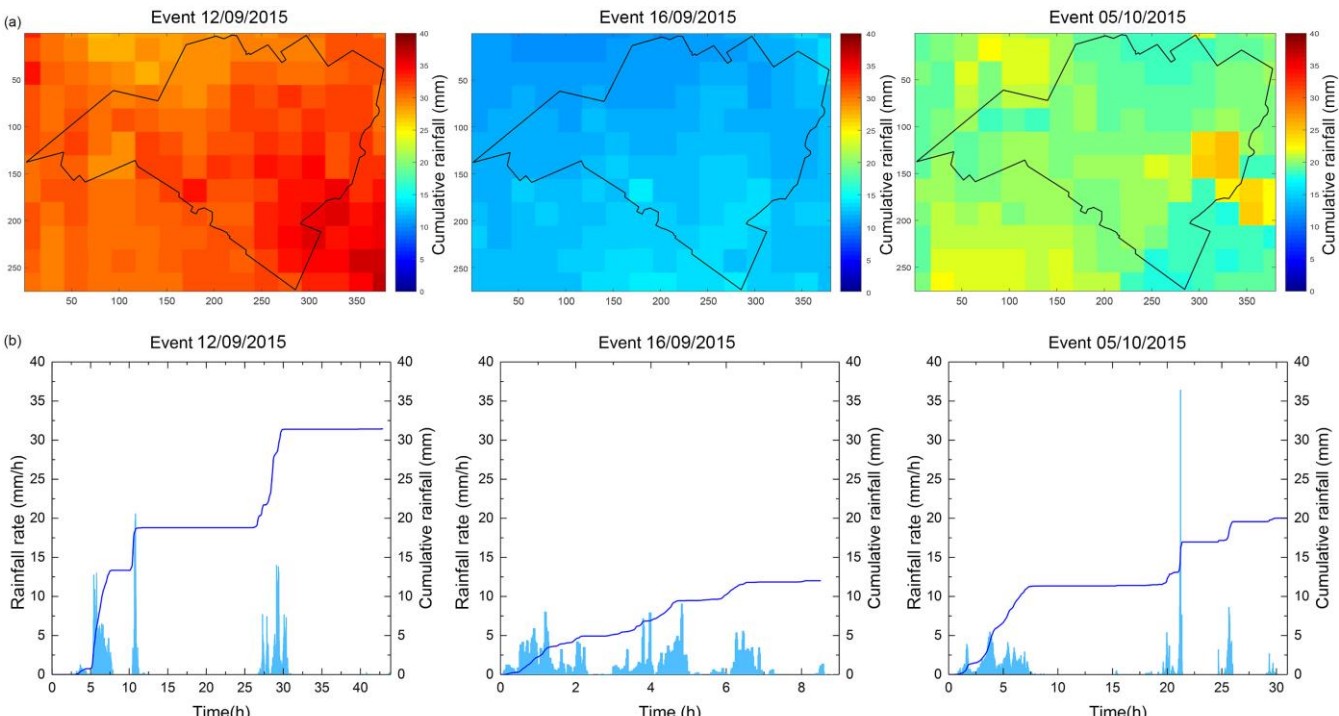

**Figure 2: (a) Cumulative rainfall (mm) by radar pixels of three rainfall events, (b) time evolution of rainfall rate (mm/h) and cumulative rainfall (mm) of the three rainfall events over the whole catchment.**

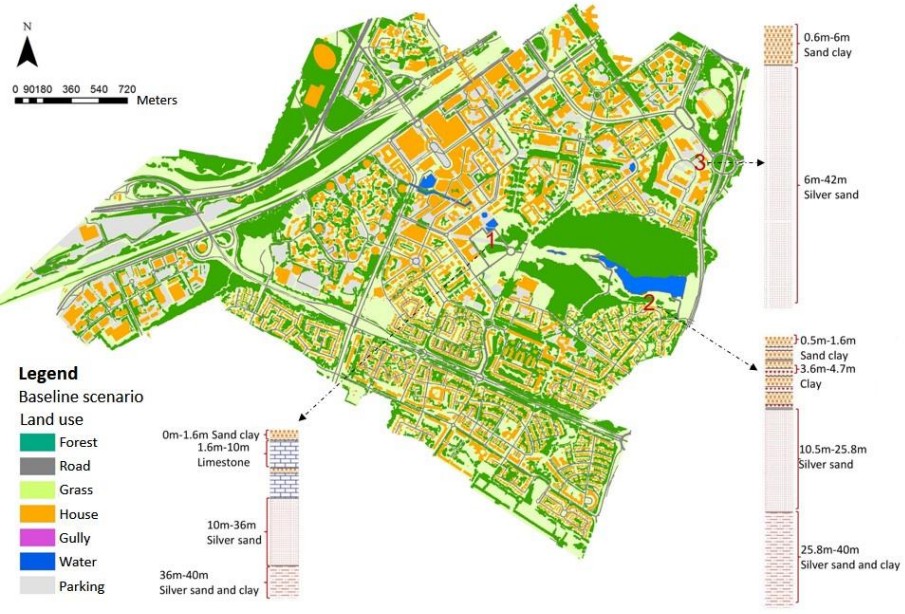

**Figure 3: Land use map and soil data.**





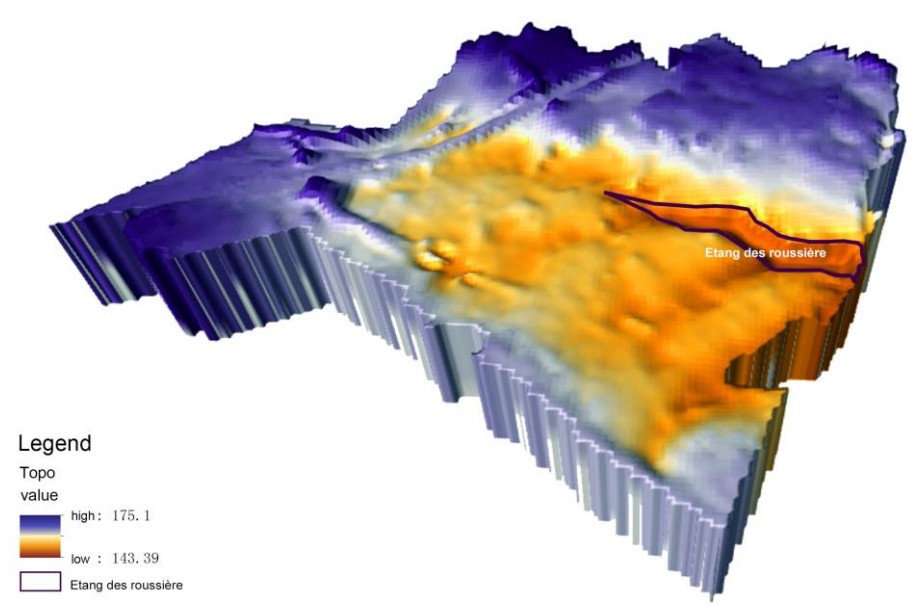


**Figure 4: 3D Topography - 25 m resolution DEM.**

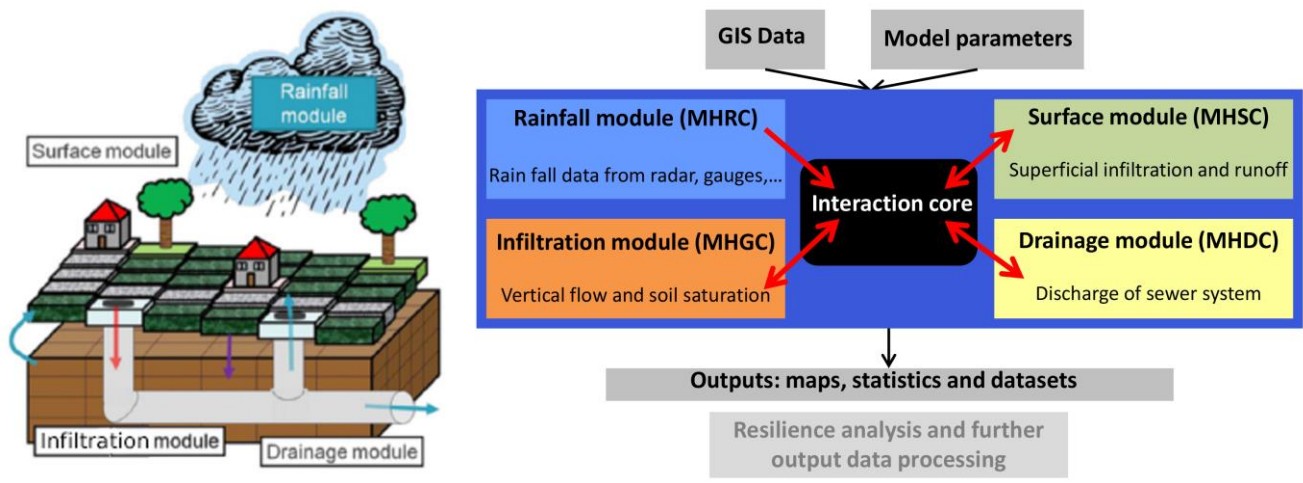

**Figure 5: The framework of Multi-Hydro model.**



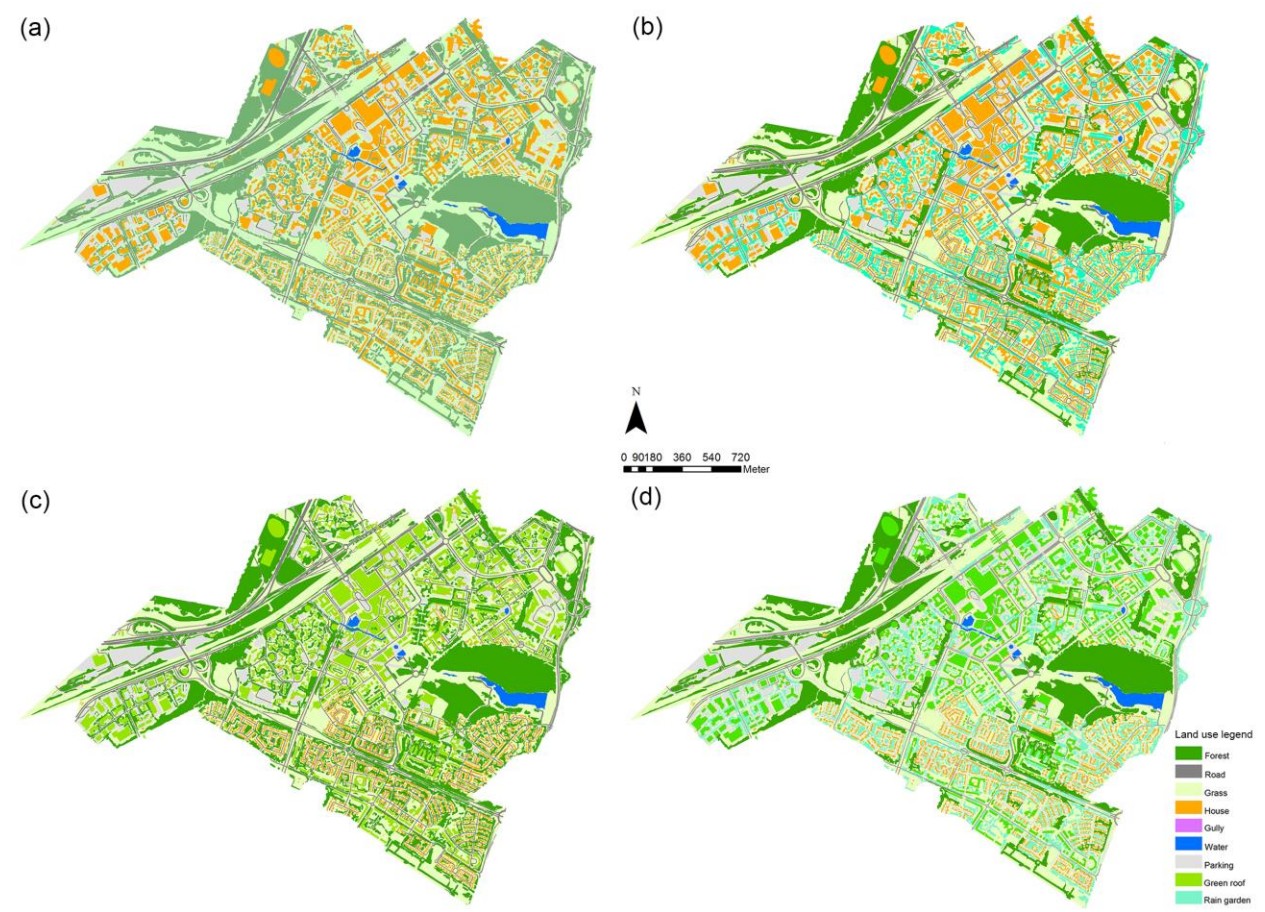

**Figure 6: LID/NBS scenario: (a) porous pavement scenario, (b) rain garden scenario, (c) green roof scenario, (d) combined scenario**

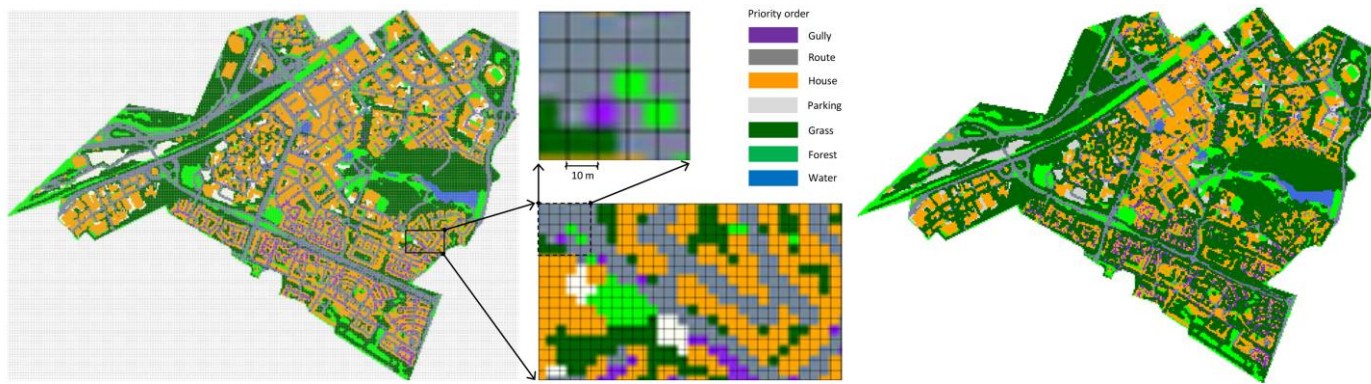

**Figure 7: Raster land use map with 10 m spatial resolution: priority rule (left), majority rule (right).**





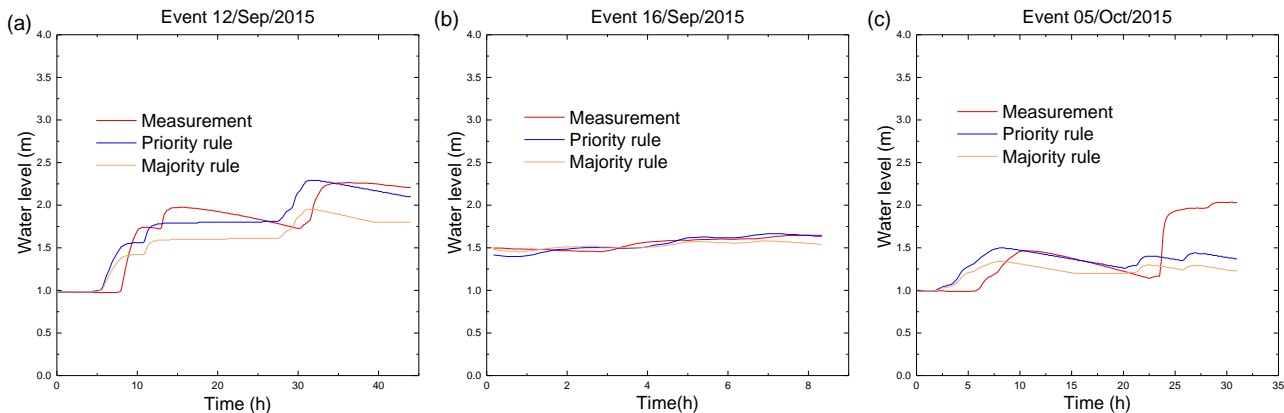

**Figure 8: Comparison of observed and simulated water level (land use with priority rule and majority rule) of three rainfall events: (a) 12/092015; (b) 16/09/2015 and (c) 05/10/2015.**

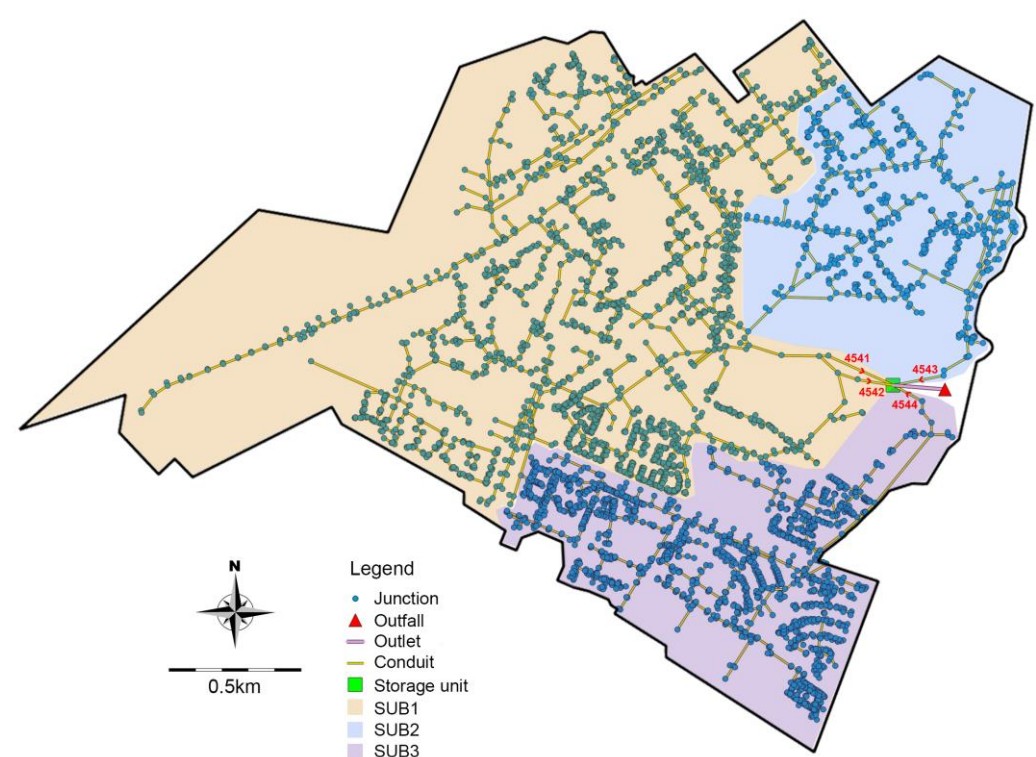

**Figure 9: Drainage system with four conduits (4541, 4542, 4543, and 4544) and three sub-catchments highlighted.**






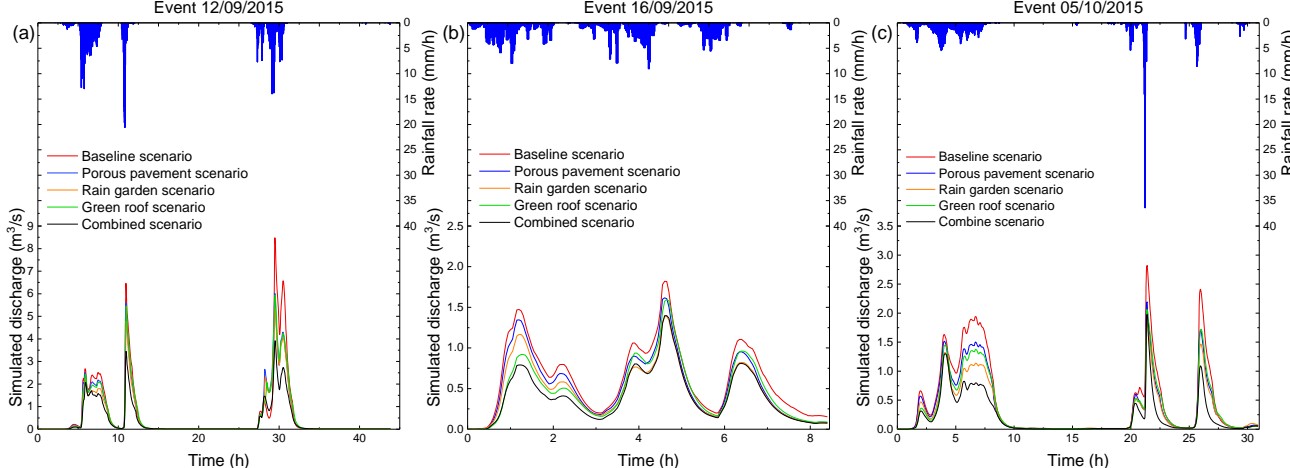

**Figure 10: Presentation of the simulated hydrographs (a, b, c) in the whole catchment for three rainfall events and five scenarios.**




**Figure 11: Presentation of the simulated hydrographs (a to i) in three sub-catchments for three rainfall events and five scenarios.**





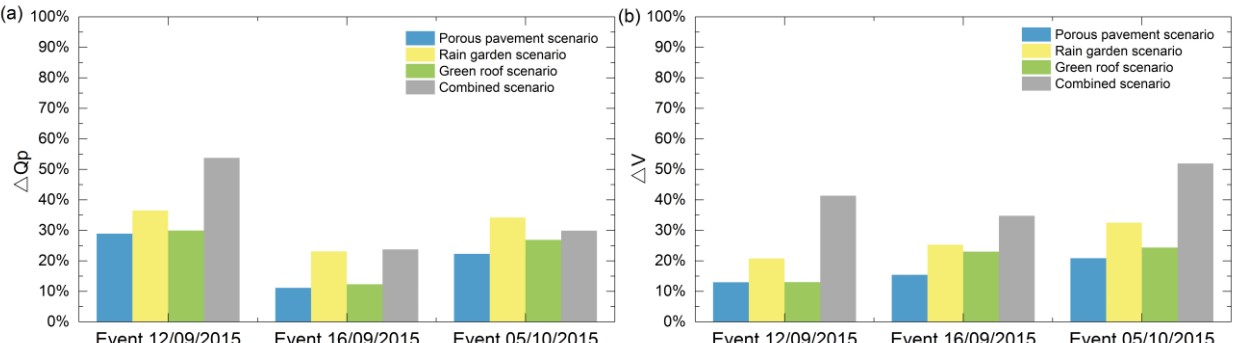

**Figure 12: Percentage of total runoff volume reduction and peak discharge reduction of four LID/NBS scenarios, compared with baseline scenario in the whole catchment.**

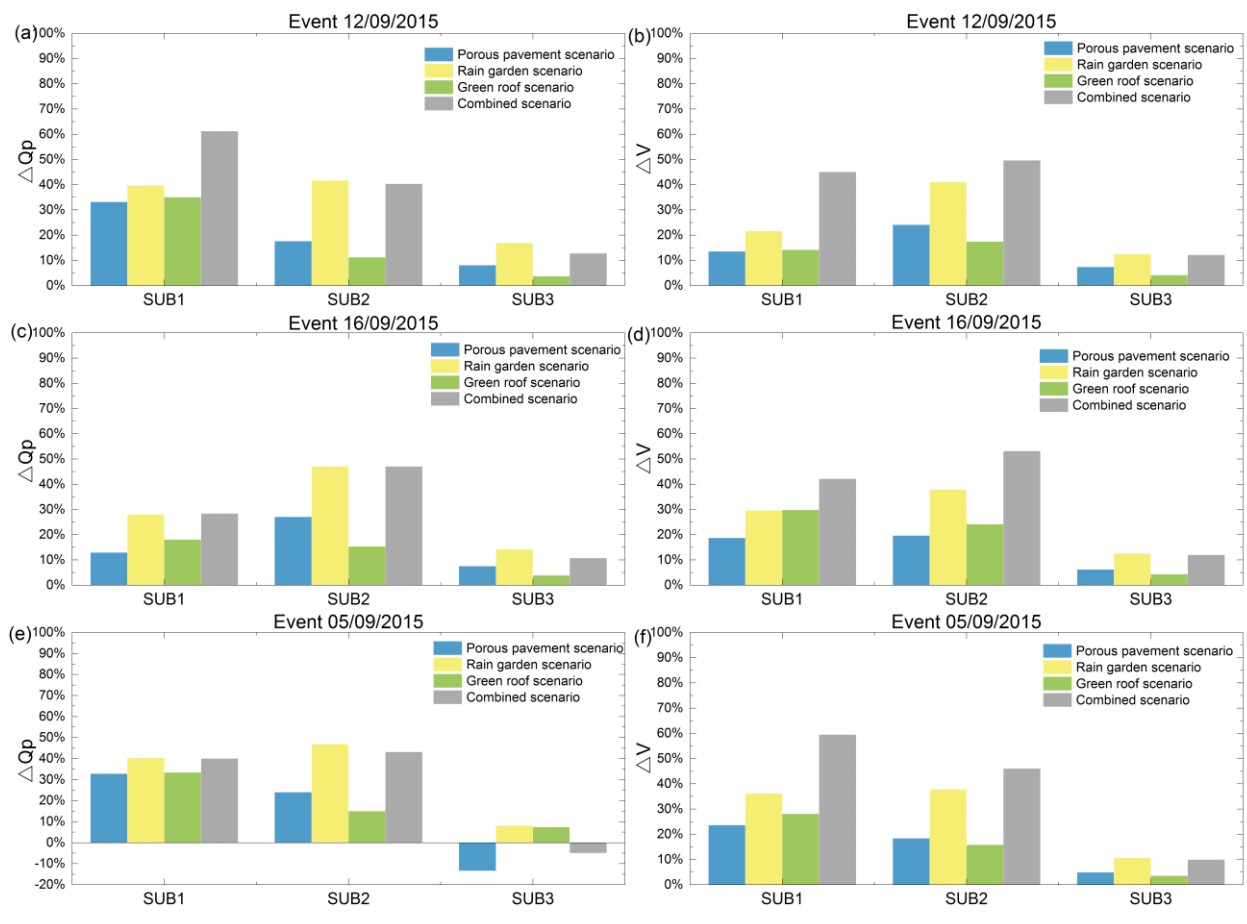


**Figure 13: Percentage of total runoff volume reduction and peak discharge reduction of four LID/NBS scenarios, compared with baseline scenario in three sub-catchments.**





**Table 1. Characteristics of three rainfall events.**

| Date | Cumulative rainfall (mm) | Duration (h) | Return period |
|---|---|---|---|
| 12-13/09/2015 | 31.4 | 44 | 6 months |
| 16/09/2015 | 12.0 | 16.8 | 1 months |
| 05-06/10/2015 | 20.0 | 31 | 2-5 years |

**Table 2. Hydrological parameters for each land use.**

| Land use | Hydraulic conductivity (m/s) | Manning's coefficient (no units) | Interception (mm) |
|---|---|---|---|
| Grass | 1.9e-6 | 0.15 | 3.81 |
| Forest | 1.9e-6 | 0.8 | 7.62 |
| Roads | 1.0e-10 | 0.012 | 1.9 |
| Houses | 1.0e-10 | 0.012 | 1.9 |
| Gullies | 1.0e-0 | 0.9 | 0 |
| Parking | 1.0e-10 | 0.012 | 1.9 |
| Water | 1.0e-0 | 0.9 | 100 |
| Porous pavement | 1.0e-4 | 0.014 | 2.14 |
| Rain garden | 1.9e-5 | 0.2 | 7.62 |
| Green roof | 3.3e-4 | 0.14 | 3.81 |

**Table 3. Soil properties from the VS2DT software.**

| Soil type | Kh/Ks (no unit) | Ks (m/h) | Porosity (no unit) | Alpha (m-1) | Residual moisture (no unit) | Beta (no unit) |
|---|---|---|---|---|---|---|
| Sandy clay | 1 | 0.0012 | 0.38 | 0.1 | 2.7 | 1.23 |
| Silver sand | 1 | 0.0875 | 0.377 | 0.072 | 1.04 | 6.9 |
| Silver sand and clay | 1 | 0.002 | 0.38 | 0.068 | 0.8 | 1.09 |




**Table 4. Green roof properties from the Multi-Hydro model manual.**

| Soil thickness (m) | Hydraulic conductivity (m/h) | Porosity | Field capacity | Initial substrate saturation |
|---|---|---|---|---|
| 0.03 | 1.2 | 0.5 | 0.3 | 0.1 |


**Table 5. NSE coefficients and RE values of three rainfall events (priority rule and majority rule).**

| Rainfall event | NSE (priority rule) | NSE (majority rule) | RE (priority rule) | RE (majority rule) |
|---|---|---|---|---|
| 12/09/2015 | 0.99 | 0.97 | 6 % | 14 % |
| 16/09/2015 | 0.99 | 0.99 | 2.2 % | 2.8 % |
| 05/10/2015 | 0.95 | 0.93 | 15 % | 18 % |


**Table 6. The proportion of three kinds of LID/NBS practices and the average cumulative rainfall in three sub-catchments.**

| Sub-catchment | Average cumulative rainfall (mm) | | | % of LID/NBS practices | | | |
|---|---|---|---|---|---|---|---|
| | Event 12/09/2015 | Event 16/09/2015 | Event 05/10/2015 | Porous pavement scenario | Rain garden scenario | Green roof scenario | Combined scenario |
| SUB1 | 29.7 | 11.1 | 19.6 | 9 % | 5 % | 10.9 % | 24.9 % |
| SUB2 | 32.1 | 11.3 | 19.6 | 5 % | 7.4 % | 10.3 % | 22.7 % |
| SUB3 | 33.5 | 13.1 | 20.1 | 4% | 12% | 3.5% | 19.5% |