# Peer review of "Evaluation of Low Impact Development and Nature-Based Solutions for stormwater management: a fully distributed modelling approach"

_Hydrology and Earth System Sciences, 2019_

## Referee Comment (RC1) · Anonymous Referee #1 · 2 Sep 2019

The manuscript presents a sustainable approach for storm management, which is interesting to the hydrologists and decision-makers. However, I do not see a novel contribution, the manuscript is near to be consultant report rather than academic research. The manuscript does not introduce new method or concept. In addition, the application has been applied for only one case study and show the findings while such findings could be completely difference in case it has been applied in different case study.

---

## Author Comment (AC1) · 5 Sep 2019

Dear Referee,

We first would like to highly acknowledge your prompt actions after accepting to review our manuscript (No.hess-2019-347). However, we are quite surprised by the generality of your brief comment on it and by lack of any specific comment or suggestion, in particular constructive ones. For instance, we would have appreciated to have some indications on what in the presentation of our paper could have helped you believe that it is "near to be consultant report rather than academic research". It seems that you overlooked our statement that "most studies on modelling the hydrological impacts

of LID/NBS practices are based on semi-distributed hydrological models and design storms", whereas our paper has the (academic) originality to focus on the coupling effects of variability of spatial distributions of precipitation and land uses by a fully distributed modelling approach, as well as with the help of high resolution radar data. This paradigm shift from homogeneous modelling and data to extremely heterogeneous ones is achieved with the help of the fully distributed hydrological model (Multi-Hydro). It enables us to investigate the hydrological performances of LID/NBS practices by using three sets of high resolution distributed rainfall data from ENPC X-band radar as the meteorological inputs. The Multi-Hydro model considers the land uses at 10 m scale (each land use class was presented by a 10 m x 10 m pixel, and the whole catchment includes 104500 pixels in total), which far exceeds the limitation of semi-distributed hydrological model in reflecting complicated urban catchments (e.g., SWMM averages land uses and precipitations for each sub-catchment (Burszta-Adamiak and Mrowiec, 2013, Rossman et al. 2010). The spatial resolution of ENPC X-band radar data is 250 m, which reflects the spatial variability of the precipitation at a very high resolution. To the knowledge of the authors, this is the first time that high resolution X-band radar data applied for evaluating the hydrological performances of LID/NBS practices. The other researches usually use homogeneous rainfall, which did not consider the rainfall variability in space (Ahiablame and Shakya, 2016; Hu et al., 2017; Bloorchian et al., 2016; Zahmatkesh et al., 2015; Sun et al., 2014; Qin et al., 2013; Peng et al., 2019). Last but not the least, we also find a new way of presenting the results that makes their interpretation much more general and easily scalable for other studies.

Obviously, we will be happy to seriously take into account any concrete suggestion which helps improve our manuscript.

---

## Referee Comment (RC2) · Anonymous Referee #1 · 6 Sep 2019

In fact, I do not need to give specific comments on a report "from my point of view", that I do not see any specific new knowledge to the readers. Using an existing model with different input data pattern is NOT considered as a new idea, a novelty in knowledge or new observations. The statement that you claimed "I overlooked it", the authors used a ready-made model, the authors did not even make any modification in the model. Finally, I recommend the authors to try to apply the same concept in another case study, and hence, the findings from both case studies could lead the authors to acquire or observe a new knowledge that might be useful and valuable for the hydrologists.

In addition, the authors might carry out a lot of contribution but it is not clear in the current version of the paper. This reviewer appreciates so much if the authors could improve the current version of the manuscript by focusing on their contribution by adding the real details of modeling structure that might be new knowledge to the hydrologists and academic society.

---

## Referee Comment (RC3) · Anonymous Referee #2 · 9 Sep 2019

General comments

The paper describes how a hydrological model is used to forecast the effect of implementing a number of LID/NSB practices in a model of an urban area. Such studies have been performed before (as reviewed in the introduction). The authors identify two shortcomings of such previous studies, namely (1) that they do not (or only in a limited way) consider the spatial variation of land covers in urban areas, and (2) that they do not consider the spatial variation of rainfall. The study then presents a model setup with distributed land cover information and spatially distributed rainfall, and finds that changes in the land cover (implementing porous pavements, rain gardens, or green
roofs, or combining all three of these) result in reduced runoff rates and volumes.

My primary concern with the paper is that it does not present a new contribution to scientific progress. The current paper only uses a somewhat more complex model to arrive at similar conclusions as earlier papers. The paper does not actually address what the effect of using a fully distributed modelling approach is, in relation to the two shortcomings of non-distributed approaches identified in the introduction. (E.g. does the fully distributed model give significantly different results than other approaches? If differences are found, can these be attributed to the distributed land cover data or the distributed rainfall data? Is the magnitude of the differences between the methods so large that it justifies the use of the more demanding (in terms of data and user effort) fully distributed approach, in relation to the uncertainties that will exist in any forecast?) For the model to be a useful forecasting tool, it would need to be shown that it provides more accurate and precise forecasts than simpler methods. The paper does show that results vary between three different subcatchments; however, since these subcatchments appear to be 1-3 km2 large (if the exact sizes are given I missed them), this does not help demonstrate any added value of a model with a much finer (10 m) discretization.

The three main conclusions of the paper do not seem useful or valid to me:

1. "The results illustrate that implement LID/NBS practices can significantly reduce the urban runoff." This has been reported many times before, and it is unsurprising, given that this is the behaviour that would be expected of the model (and in fact this is what LID facilities in models are designed for), so this is not a particularly interesting finding either. The forecast effects of the LID measures are not validated against any measured data. Although I understand that this would be difficult to do, this does not change the fact that, without such validation, it cannot be judged whether the proposed approach is actually a useful forecasting tool.

2. "In the whole catchment, each LID/NBS scenario is more effective in two stronger

but short events." I do not understand what is meant here.

3. "In the sub-catchments, the significantly different hydrological responses of LID/NBS scenarios indicate that their performance is influenced by the coupling effect of variability of spatial distributions of precipitation and land uses (e.g., the rainfall amount, rainfall intensity, proportion of LID/NBS practice)." I do not think this is supported by the results of the paper. It seems obvious to me that different subcatchments with different characteristics will respond differently to the implementation of LID measures. The effect of representation of spatial variability of land cover and precipitation is not actually tested in the paper.

To me, it does not seem feasible to address the issues above in a modified version of the manuscript, as it would rather require a whole new study. (As outlined below, there are also some more specific shortcomings/question surrounding the chosen methodology.) Therefore I recommend this manuscript be rejected by HESS.

Specific comments

The manuscript would require good language editing, as it currently contains many grammatical errors.

L72-73: if some research does use more detailed data (as this sentence states) then this is the most relevant literature to be reviewed in the introduction, yet no references are given!

L127: the resolution of the DEM is coarser than that of the model. This limits the value of having the model at that resolution, and it may also lead to problems with the surface runoff module. Is there no higher resolution DEM available?

L133-138: Using only three sampling points for soil classification may be too limited. Although it may be appropriate for the deeper soil layers, studies have shown that urban areas have a high degree of spatial variability in the top layer of the soil and/or the infiltration capacity. Combining a fully-distributed model with uniform data runs the

risk of getting the worst of two worlds, i.e. lots of work to set up the model, but not actually using more information than coarser modelling approaches.

L163-L178: the proposed scenarios assume that LID measures are implemented on all suitable surfaces, is this realistic? When LIDs are assumed to be implemented on all usable surfaces, the comparison between their effects on runoff may not be that useful, since other relevant factors (e.g. installation cost, operating cost, social acceptance, other physical limitations) can be different for the different LID scenarios.

L172-L175: although steep roofs are typically unsuitable for green roofs, green roofs may have gentler slopes.

L180-L186: The proposed modelling approach is rather detailed, but it is evaluated only based on peak flow and total runoff volume. Is such a detailed modelling approach really needed/justified if these rather simple metrics are the quantity of interest? Alternatively, are there other relevant metrics where the benefit of the detailed approach would be clearer?

L204-L206: although the non-calibrated model appears to function for the first two events (figure 8), there is a major deviation for the third event. Wouldn't it be better to calibrate the model further so that it better simulates the catchment behaviour? After all, the model being accurate for the current situation is a fundamental requirement for putting faith on its forecasts of the effect of changes in the system. The NSE values for the third event should be checked as (based on the graph) they appear to be quite high given the large deviation during the last 20-25% of the event duration.

L212: A model result with a NSE close to 0 is not "credible", see e.g. Moriasi et al (2007). After all, 0 is the score that would be achieved by the average of the observations, which is not a particularly strict (albeit a commonly used) benchmark.

Moriasi, D. N., J. G. Arnold, M. W. Van Liew, R. L. Bingner, R. D. Harmel, and T. L. Veith. 'Model Evaluation Guidelines for Systematic Quantification of Accuracy in

[Figure]

Watershed Simulations'. Transactions of the ASABE 50, no. 3 (2007): 885–900. https://doi.org/10.13031/2013.23153.

Technical corrections

The scale of the y-axis in the graphs leaves a lot of empty space. It would be better to use this space to show the data in more detail, as it is now difficult to see differences between the different lines.

Given the major shortcomings of the paper outlined above, I will refrain from spending too much time on small issues here.

---

## Referee Comment (RC4) · Anonymous Referee #3 · 13 Sep 2019

The manuscript tries to answer the question regarding the effect of nature-based solutions and low impact development. The manuscript use of English is not sufficient and the flow and structure of the sentences and paragraphs make it very harder to truly review this manuscript. Moreover, I see neither scientific merit nor engineering merit in this manuscript. I cannot even assess the quality of this manuscript as a technical memorandum.

There are a lot to be mentioned for this manuscript.

1- Uncertainty of the model, the simulations, model structure, X-band radar, downscaling etc etc. . . the values and ratio, how can 0.1 cms change be really evaluated given

the total discharge.

2- Validity of the model

3- Feasibility of the nature-based low impact study (it is feasible to have so much change in this region. . . how much does it cost?!)

4- Clear research question/real engineering purpose and problem-solving.

5- Context is missing. Lots of work done on water sensitive cities from years ago. Also, similar works have been done using X-band radar (at least I know of Rotterdam and I am sure it should be more cases around the world).

6- Where is the novelty of this work? and what did I learn when reading this work? I have done work on LID myself so I am familiar with the development but this work is not really adding to what I knew...

Unfortunately, I think the manuscript is not suitable for HESS from presentation point of view, engineering point of view and scientific point of view.

---

## Author Comment (AC2) · 13 Sep 2019

Dear Referee,

We appreciate receiving more concrete comments. However, we do not see any link to our paper with the general comment "Using an existing model with different input data pattern is NOT considered as a new idea, a novelty in knowledge or new observations". Indeed, none of these were goals of our paper. Still, the main goal of our paper seems to remain "overlooked": the paradigm shift from homogeneous modelling and data to extremely heterogeneous ones, as put forward in our previous reply.

[Figure]

On the contrary, we agree that we have to revise our paper to:

- put more in evidence the aforementioned general methodological goal

- include a comparison of homogeneous runs (project rainfall) vs. heterogeneous runs to demonstrate limitations of the homogeneous modelling (we indeed hesitated to include preliminary results in the first version)

- make more obvious that we have been working on three sub-catchments, rather than a unique one, as well as with a given variety of scenarios

- underline that this diversity reinforces the validity of our conclusions with respect to heterogeneity

- be more specific on the necessary adaptations of the hydrological Multi-Hydro model to include LID/NBS solutions, as well as the fact that all authors have contributed to the ongoing development of this model (i.e. it is not taken from a shelf).

Incidentally, this exchange could confirm the interest of interactive comments between authors and referees, especially with respect to the classical, one-way review reports.

---

## Author Comment (AC3) · 17 Sep 2019

Dear referee #2,

The authors would like to thank you for your comments on our manuscript. Your first general comment acknowledges the originality of our paper, i.e. that is focused on the spatial variability of both land cover and rainfall to assess/forecast the performance of LID/NSB. On the contrary, we have some difficulty to understand how you can then feel concerned that our manuscript "does not present a new contribution to scientific progress"! For instance, confirmation of previous results obtained with simpler models would already be meaningful. However, this was neither the aim nor a result of our

paper focused on showing the space-time complexity of the basin response, as it can be inferred from high space-time resolution data and modelling. Although it was not one of our original aim, we agree that our manuscript should be substantially revised to make more obvious the gains obtained with a fully distributed modelling approach is, especially with respect to the shortcomings of non-distributed approaches that we identified. In fact, we already obtained direct comparisons between both approaches, but they were not included in the present version of our manuscript for the aforementioned reasons. But, there is no fundamental difficulty to include these comparisons in the revised version.

Please, see in the attached supplementary file our point-by-point responses to the referee's specific comments and our subsequent suggestions to improve our paper.

Please also note the supplement to this comment:
https://www.hydrol-earth-syst-sci-discuss.net/hess-2019-347/hess-2019-347-AC3-supplement.pdf

**Supplement:**

Q1: The forecast effects of the LID measures are not validated against any measured data.

R1: Our paper does not deal with a unique LID/NBS measure, but gives insights on how to optimise the choice among a set of them. Obviously, it is impossible to have the means to empirically validate a set of measures on a given basin, but this does not prevent to partially validate the numerical simulations with a degree of confidence that is closely related to that of the validation of the baseline scenario. In the present case, the latter was achieved with a high degree of confidence (e.g. the NSE coefficients for the event 16/09/2015 is 0.99)

Q2: "In the whole catchment, each LID/NBS scenario is more effective in two stronger but short events." I do not understand what is meant here.

R2: The two stronger events are composed by several short but intense sub-events and some inter dry periods. The sentence means the LID/NBS practices have better performance in terms of the intermittent rainfall (sometimes intense). To our knowledge, this is a characteristic that has been largely unnoticed so far.

Q3: It seems obvious to me that different sub-catchments with different characteristics will respond differently to the implementation of LID measures. The effect of representation of spatial variability of land cover and precipitation is not actually tested in the paper.

The first sentence supports the main goal of our paper and explains its main mechanism. But, contrary to the second sentence, our paper did deal with it. As discussed above, what could have been missing is a direct comparison with low spatial variability modelling that we are willing to introduce in the revised version.

Q4: L72-73: If some research does use more detailed data (as this sentence states) then this is the most relevant literature to be reviewed in the introduction, yet no references are given!

*L72-73: In general, rare research used the high resolution rainfall data (i.e., X-band radar data with 250 m spatial resolution) and considered the coupling effect of the spatial distributions of precipitation and land uses.*
We have to confess that the word "rare" is too diplomatic and confusing. This sentence will be corrected to clearly state that to the knowledge of authors, this is the first time that X-band radar data are used as meteorological inputs to analyse the hydrological impacts of LID measures, and to consider the coupling effect of the spatial distributions of precipitation and land use.

Q5: L127: The resolution of the DEM is coarser than that of the model. This limits the value of having the model at that resolution, and it may also lead to problems with the surface runoff module. Is there no higher resolution DEM available?

In this study, the available DEM data had a 25 m resolution and they were downscaled by interpolation to the resolution of 10 m. We agree that the results could be more reliable with higher DEM resolution will enable to run Multi-Hydro with higher resolution as well, but these data are not

presently available.

Q6: L33-138: Using only three sampling points for soil classification may be too limited. Although it may be appropriate for the deeper soil layers, studies have shown that urban areas have a high degree of spatial variability in the top layer of the soil and/or the infiltration capacity. Combining a fully-distributed model with uniform data runs the risk of getting the worst of two worlds, i.e. lots of work to set up the model, but not actually using more information than coarser modelling approaches.

*L133-138: Soil data for the catchment was obtained from the InfroTerre Database (http://infoterre.brgm.fr). As shown in Fig. 3, the soil data of three points are selected, which indicates that sand clay is located at the first layer of the soil profile. For point 1, there*
*135 exists a layer of limestone soil, which is less permeable. Silver sand which has the best infiltration ability, was in the next layer. The last layer is silver sand and clay, which has better infiltration ability than limestone. From a hydrological point of view, the soil data shows the complexity of the subsurface of the catchment. Therefore, the soil profile is reasonably simplified into three layers: sand clay layer (0-10.5 m), silver sand layer (10.5-25.8 m) and silver sand and clay layer (25.8-40 m).*

In Multi-Hydro, the infiltration process is a two-stage process:
-firstly estimated with the help of the surface module based on the highly variable surface data (land use)
-secondly, if this first estimate is strong enough, this infiltration in the soil is estimated with the help of the infiltration module with possible feedbacks to the previous step estimate.
We certainly have to explain that although we are interested by the highest available resolution of all data, those of the soil have usually a much coarser resolution than the surface data, but fortunately have only a sort of second order impact. This explains why commercial softwares drastically reduced the number of available layers with respect to their initial versions.

Q7: L163-L178: The proposed scenarios assume that LID measures are implemented on all suitable surfaces, is this realistic? L172-175: Although steep roofs are typically unsuitable for green roofs, green roofs may have gentler slopes.

*L163-L178 : For rain garden scenario (Fig. 6b), the low elevation greenbelts around houses were implemented by rain gardens, which can collect and store up the surface runoff from surrounding impermeable areas before infiltration on site. When rain garden saturated, the redundant surface runoff will drain into the drainage system. On the basis of application condition of rain gardens and the urban planning of the city of Guyancourt, 11.5 % of the whole area is set as rain gardens in the catchment. In the catchment, most of the buildings are houses with sloped roofs. Other types of buildings with flat roofs, only constitute one third of the total building area. According to the properties of green roof, small and light green roofs consisting of a soil layer and a storage layer are implemented on all flat ones, which can be simulated by the green roof module. All slope roofs remain unchanged. Finally, green roofs were applied to 11.5 % of the whole area (Fig. 6c). The combined scenario (see Fig. 6d), combined the three aforementioned LID/NBS practices. Those practices occupy 37.5 % of the whole catchment. In this case, the area of pervious surface reached*

*4.6 km2, which is about 88.4 % of the whole catchment.*

Indeed, the proposed scenarios were based on a maximal implementation with respect to given physical properties of LID measures and site conditions. There is no difficulty to (randomly) select only a proportion of them to take into account other factors such as costs, social acceptance etc., as well as to modify site conditions, e.g. to enlarge the potential green roofs by including roofs with gentle slope.

Q8: L180-186: Evaluated only based on peak flow and total runoff volume, alternatively, are there other relevant metrics where the benefit of the detailed approach would be clearer?

*L180-186: The general hydrological response of five scenarios under three rainfall events has been assessed by Multi-Hydro model and its green roof module (3×5 simulations). Two index (ΔV, total runoff volume reduction; ΔQp, peak discharge reduction) are calculated:*

$$\Delta Qp(\%) = \frac{Qp_0 - Qp_i}{Qp_0} \times 100$$

$$\Delta V(\%) = \frac{V_0 - V_i}{V_0} \times 100$$

*where $Qp_0$ refers to peak discharge and $V_0$ represents total runoff volume in baseline scenario, and the $Qp_i$ and $V_i$ are peak discharge and total runoff volume of other scenarios with different LID/NBS.*

Certainly, there are other metrics to better highlight the space-time heterogeneity of the basin response and we are currently testing some of them to be included in the revised version, e.g. local ΔQp(%) and ΔV(%), as well as their quantiles. However, the peak flow and total runoff volume are presumably the most suitable metrics to make the comparison of the results of homogeneous and distributed data and modelling.

Q9: L204-206: There is a major deviation for the third event. Wouldn't it be better to calibrate the model further so that it better simulates the catchment behaviour?

*L204-206 All the model parameters related to the land use type and soil type were selected from the Multi-Hydro model manual (Giangola-Murzyn et al., 2014), as shown in Table 2 and Table 3.Green roof is a special LID/NBS practice which needs to be simulated with the Multi-Hydro green roof module. The properties of the green roof are illustrated in Table 4.*

Multi-Hydro being physically-based, does not require calibration, contrary to conceptual models and the expression "model parameters" is somewhat misleading. It should be replaced by physical constants (e.g. conductivity) defined by land use and soil types.
The major deviation of the third rainfall events could probably be due to that the highest rainfall peak of radar data only last 3 min, and the simulated retention pond couldn't accumulate enough water compare to the measurement data. We simulated the same rainfall event with the C-band radar data with the spatial resolution of 1 km, and temporal resolution of 5min. For the validation results of the C-band radar data, the major deviation is less obvious than that of the results of X-

band radar (see additional Fig.8). This could be explained by the fact that C-band radar is closer to the catchment and could better detect the rainfall intensity. Our co-authors indicated (Paz.et al, 2018) that the X-band radar tends to underestimate the total accumulated rainfall depth.

[Figure]

Fig.8 (additional) Comparison of observed and simulated water level with two different type of radar data (X-band and C-band) of event 05/10/2015

Q10: L212: A model result with a NSE close to 0 is not "credible", see e.g. Moriasi et al (2007). After all, 0 is the score that would be achieved by the average of the observations, which is not a particularly strict (albeit a commonly used) benchmark.

*L213-214-The Nash is closer to 1, indicating that the model is more reliable. Nash is closer to 0, indicating that the simulation result is closer to the average observed value, which means the result is credible.*

We apologise for the typo in L214: "which means the result is credible" should indeed read "which means the result is not credible"

Thanks to the helpful reference.

Q11: The scale of the y-axis in the graphs leaves a lot of empty space. It would be better to use this space to show the data in more detail, as it is now difficult to see differences between the different lines.

We modified the graph according to your suggestion. The modified graphs are attached.

[Figure]

Figure 11(additional): Presentation of the simulated hydrographs (a to i) in three sub-catchments for three rainfall events and five scenarios.

---

## Referee Comment (RC5) · Anonymous Referee #1 · 18 Sep 2019

I really do not understand your way in answering my comments. If I DID NOT UNDER-STAND and usually "OVERLOOK" the goals of your works, it means that you can not present the goal of your work in a proper way and you should try and work harder to let your goals clear for the readers.

"On the contrary, we agree that we have to revise our paper to", this statement is very strange, if you know that you need to revise your paper considering the comments "you added in your reply after this strange statement", my comment is (Why you did NOT do that before you submitted your paper), I hope that this comment is clear for you.

When I asked you to follow the same procedure "your current work from A to Z" to

be applied in another case study due to the fact you might find out that the drawn conclusion might be different in the other case study.

You applied a certain approach in a particular case study and achieve almost well-known outcomes, NOW, my clear comment is to apply this procedure in a different case study. Definitely, the case study should be different than the one you presented in your current work, show the results, discuss the findings and present that clearly in a revised version of this work.

---

## Author Comment (AC5) · 18 Sep 2019

Dear referee #3,

We would like to thank you for your interactive comment.

We noticed that your short summary of our paper does not mention at all to that our evaluation LID/NBS effects is based on a fully distributed modelling approach, whereas we believe is a research novelty. This may explain why you did not see any scientific merit to our paper, which obviously has nothing to do with a technical memorandum. We are willing to improve the English of our manuscript. We also confess not being

familiar with expressions like "very harder".

Below you will find our point-by-point answers to your comments:

Q1- Uncertainty of the model, the simulations, model structure, X-band radar, downscaling etc etc... the values and ratio, how can 0.1 cms change be really evaluated given the total discharge.

Our paper did not aim to give a detailed presentation of Multi-Hydro and multifractal downscaling. However, the section 2.3 provides a synthetic presentation of them with the support of a dozen of references to detailed presentations, as well as with some more specialised sessions (e.g. Sect. 2.6 "Modelling set-up"). Most your questions are discussed there, particularly the modular structure of Multi-Hydro and the fact that Multi-Hydro has been validated on many catchments by the co-authors. Furthermore, our paper presents a detailed and successful validation of the baseline scenario on the studied catchment (Sect. 3.1).

Firstly, the 0.1 cms can absolutely be simulated by Multi-Hydro, whose accuracy is better than $1.5 \times 10^{-4}$ cms.

Secondly, the 0.1 cms discharge difference between the porous pavement scenario and baseline scenario at the highest rainfall peak of event 05/10/2015 in the sub-catchment (SUB3) is small, but nevertheless represents about 20% of the peak of the baseline scenario and not negligible with respect to those of the largest sub-catchment SUB1 and of the whole catchment. Indeed, they involve factors in the range of 4-7 (see more details in Fig.1, enclosed).

Q2- Validity of the model

We already mentioned that this model has been validated on various basins (see references in Sect. 2.3), as well as on the baseline scenario (Sect. 3.1).

Q3- Feasibility of the nature-based low impact study (it is feasible to have so much change in this region...how much does it cost?!)

Our paper being focused on research questions, we were interested to analyse large LID/NBS implementation to possibly have largely modified basin responses. Scenarios were therefore based on a maximal implementation with respect to given physical properties of LID/NBS and site conditions. There is no difficulty to (randomly) select only a proportion of them to take into account other factors such as costs, social acceptance etc.,

Q4- Context is missing. Lots of work done on water sensitive cities from years ago. Also, similar works have been done using X-band radar (at least I know of Rotterdam and I am sure it should be more cases around the world).

We will better highlight that the originality of our paper is to be focused on space-time variability of both data and modelling. We are surprised by the statement that "similar works have been done using X-band radar" without any publication reference. Furthermore, the claim to be sure about the Rotterdam radar surprises us a lot, because our group, as partner of the EU RainGain project, is well aware of the successes and operational difficulties met by this radar.

Anyhow, we will greatly appreciate receiving references on the use of X-band radar data for analysing the performance of LID/NBS scenarios.
* * *
[Figure]

Figure 1: Presentation of the simulated hydrographs in whole catchment and in three sub-catchments (a: whole catchment, b: SUB1, c: SUB2, d: SUB3) for the scenario of baseline and porous pavement for the event 05/10/2015.

**Fig. 1.**

---

## Author Comment (AC6) · 20 Sep 2019

Dear referee,

The authors would like to thank you for your interactive comments that have very helpful to understand how to better present our results and make them more comprehensible for anyone. This is indeed a goal of the interactive discussion process.

We are willing to follow your suggestions to add the Massy case study, which is rather different with respect to the present catchment. It would indeed strengthen our conclusions.

---

## Author Comment (AC8) · 10 Oct 2019

Dear referees,

The authors would like to thank you for your helpful interactive comments.

We responded point by point to your detailed comments (see AC2, AC7, AC4_supplement, AC5). We accepted almost all of them and explained how we will take them into account in the revision of our manuscript. In the very few exceptional cases of disagreement, we clarified the probable source of misunderstandings (e.g. AC5 about the Rotterdam radar). In what follows, we summarise the main suggested

revisions.

First of all, we must emphasise the originality of our manuscript, because it deals with the NBS performance evaluation by taking into account the coupling effects of spatial variability of rainfall and land use through a fully distributed modelling approach, as well as the high resolution of X-band radar data. This is a paradigm shift from homogeneous modelling and data to extremely heterogeneous ones.

Furthermore, we also have to:

- put more in evidence the aforementioned general methodological goal, including a comparison of homogeneous runs (project rainfall) vs. heterogeneous runs to demonstrate limitations of the homogeneous modelling (see AC2 and AC4-Supplement)

- make more obvious that we have been working on non-homogeneous three sub-catchments, rather than a unique catchment

- extend our study to another catchment with very different characteristics (see AC7)

- introduce new NBS scenarios (e.g. slightly sloped green roofs) to reinforce the validity of our conclusions regarding heterogeneity (see AC4-Supplement)

- give more specific indications on the necessary adaptations of the hydrological Multi-Hydro model to implement NBS measures (see AC2)

In fact, we already obtained results corresponding to the above-mentioned modifications, and there is no fundamental difficulty to include these new findings in the revised version. With these modifications, we believe that the contributions of our paper to the advancement of hydrological modelling in applied research will be much more visible, and in agreement with the aim and scope of the Journal of Hydrology and Earth System Sciences.

We will also carefully check the quality of English of the revised version.

Finally, we would like to thank you for the time and effort devoted to review our

manuscript.